# Self-Flocculation as an Efficient Method to Harvest Microalgae: A Mini-Review

**Tianrui Li, Jiangjun Hu and Liandong Zhu ***

Hubei Key Laboratory of Biomass-Resources Chemistry and Environmental Biotechnology,
School of Resources & Environmental Science, Hubei International Scientific and Technological Cooperation
Base of Sustainable Resource and Energy, Wuhan University, Wuhan 430079, China; ltr3456@163.com (T.L.);
jjhu1963@outlook.com (J.H.)
**\*** Correspondence: author: ldzhu@whu.edu.cn

**Abstract:** The development of clean and renewable biofuels has been of wide concern on the topic of energy and environmental issues. As a kind of biomass energy with great application prospects, microalgae have many advantages and are used in the fields of environmental protection and biofuels as well as food or feed production for humans and animals. However, the high cost of microalgae harvesting is the main bottleneck of industrial production on a large scale. Self-flocculation is a cost-efficient and promising method for harvesting microalgal biomass. This article briefly describes the current commonly used technology for microalgae harvesting, focusing on the research progress of self-flocculation. This article explores the relative mechanisms and influencing factors of self-flocculation and discusses a proposal for the integration of algae cultivation and harvesting as well as the co-cultivation of algae and bacteria in an effort to provide a reference for microalgae harvesting with high efficiency and low cost.

**Keywords:** microalgae harvesting; self-flocculation; bioflocculation; extracellular polymer





## 1. Introduction

The "World Energy Outlook 2020 Edition" issued by the International Energy Agency provides a comprehensive view of the development of the global energy system in the next few decades. Due to the outbreak of an energy crisis, countries have accelerated the transition from non-renewable fossil energy, such as oil and coal, to clean energy. Taking measures to develop clean and renewable energy has become a strategic demand related to the national economy and people's livelihood. In recent years, with the continuous improvement of national economy levels, the demand for energy from countries around the world has also increased. Traditional petroleum fuels might be depleted, whose combustion releases a large amount of greenhouse gases; these are not only harmful to humans and animals but can also cause global warming and ozone depletion. In this critical period of energy transition, in order to meet global energy demand and promote economic and ecological development in a sustainable way, the development and utilization of new renewable and environmentally friendly sources of energy has become critical for environmental protection and energy utilization.

As a renewable resource, biomass energy has attracted widespread attention due to its significant advantages such as abundant raw materials, diverse energy forms and harmless combustion process [1]. Biomass energy, together with other new energy sources such as wind, solar and nuclear energy, can provide a key driver for the global energy transition. Biodiesel, as a kind of biomass energy, is a typical "green energy" with characteristics such as wide sources and renewability and is considered to be a relatively low-carbon and economic alternative energy. Microalgae are regarded as the primary raw material for biodiesel production due to their advantages such as a wide variety of species, high production efficiency, high oil content and no occupation of arable land. Shirazi et al. [2] pre-

pared spirulina biodiesel with n-hexane as co-solvent under the condition of supercritical methanol, and the highest yield reached 99.32%.

In parallel with the energy crisis, another major crisis is environmental pollution. As one of the prominent environmental pollution problems, water pollution has caused serious harm to the production and living of human beings and the ecological environment [3]. With the continuous emergence of various new water treatment technologies, efficient and cheap ecological treatment technologies have been developed rapidly; among these is the development and application of microalgae biotechnology, which has received widespread attention. Microalgae will absorb a large amount of nutrients such as nitrogen and phosphorus during their growth and absorb $CO_2$ through photosynthesis. Microalgae can also create environmental conditions for the effective removal of residual organic matters and pathogenic microorganisms in water. In addition, microalgae also have the ability to adsorb heavy metals and other harmful substances [4]. Therefore, microalgae have great potential as a deep purification technology for sewage.

However, the large-scale cultivation of microalgae is still limited by the following characteristics: small cells, negative charge and density close to water, which places algae cells in a stable suspension state in water and is not convenient for microalgae harvesting. Microalgae harvesting processes account for 20–30% of the total cost of microalgae growth, with some even up to 50% [5,6]. Microalgae cultivation in wastewater is a promising technology for renewable energy production and wastewater treatment, but the difficulty of harvesting microalgae has always been a bottleneck, restricting its large-scale production [7].

Microalgae harvesting technologies mainly include gravity sedimentation, centrifugation, filtration, flotation and flocculation. Among various methods of harvesting microalgae, flocculation is a more economical and reliable method for treating large amounts of diluted algae liquid. Flocculation is the aggregation of algal cells by sweeping, patching, bridging and charge neutralization, with the agglomeration precipitated by different types of flocculants. Flocculants can be divided into organic flocculants, inorganic flocculants and biological flocculants.

Bio-flocculation is a process in which microalgal cells are aggregated into clusters by sweeping, trapping or bridging the viscous substances produced by microorganisms or their metabolism. Compared with other flocculation methods, the biological flocculation method is environmentally friendly and sustainable. It does not require additional chemical flocculants [8]. The research directions of bio-flocculation technology are mainly divided into three categories: microorganism and microalgae co-flocculation, microbial flocculant flocculation and microalgae cell self-flocculation. In 1988, Sukenik et al. [9] first observed the self-flocculation of microalgal cells. With the in depth study of self-flocculation, more and more microalgae with self-flocculation characteristics have been found. The self-flocculation of microalgal cells is a flocculation phenomenon caused by the secretion of flocculating substances such as glycosides or polysaccharides and other flocculating substances synthesized by microalgae in the culture process, thus adhering to adjacent algal cells [10]. Microalgal cells with self-flocculation properties can not only flocculate the same species of algal cells but also induce flocculation of other types of microalgal cells [11]. In the process of microalgae culture, species, substrate concentration, pH value, temperature, light intensity and other environmental factors will have a certain influence on the self-flocculation phenomenon of microalgae cells.

*Structure of this Work*

This paper starts by discussing the mechanism of self-flocculation based on the flocculation efficiency of self-flocculating algae species. Subsequently, it analyzes the self-flocculation influencing factors and compares the advantages and disadvantages of various biological flocculation methods. Finally, it discusses the integration of algae culture and biomass collection as well as co-culture technology of algae and bacteria in order to provide a reference for follow-up research on self-flocculation.

## 2. Self-Flocculation Mechanism and Harvesting Efficiency

### 2.1. Self-Flocculation Mechanism

Regarding the mechanism of microalgae self-flocculation, it is generally believed that microalgae self-flocculation is caused by two mechanisms [12,13].

1.  Under the circumstances of alkalinity, calcium and magnesium ions form positively charged precipitates which play an electrically neutralizing role with the negative charge of microalgae cells, thus inducing cell self-flocculation. The flocculation process is shown in Figure 1a. The high pH state can be formed naturally by the absorption of $CO_2$ ($HCO_3^-$) in the water during photosynthesis of microalgae or through the artificial addition of alkaline substances such as lime or sodium hydroxide.
2.  Some algae species can produce a large amount of extracellular polymer (EPS) during their physiological activities, and EPS act as a biological flocculant to induce flocculation. The flocculation process is shown in Figure 1b. *Actinastrum*, *Micractinium*, *Scenedesmus*, *Coelastrum*, *Pediastrum* and *Dictyosphaerium*, which are commonly found in high-efficiency algae pools, form large community structures (50–200 μm) through this mechanism and then settle naturally [14].

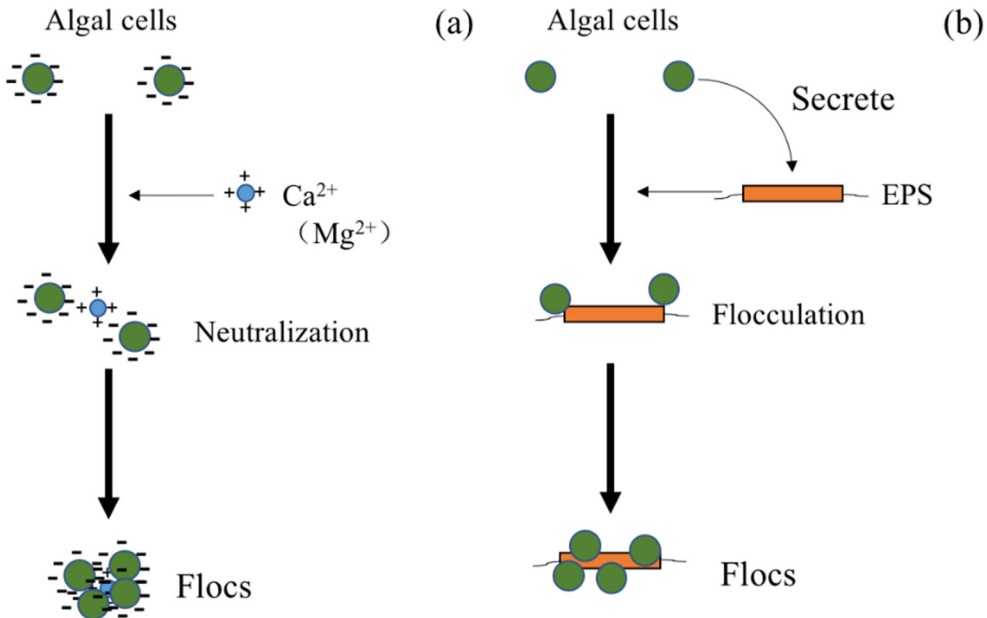

**Figure 1.** Self-flocculation mechanism of microalgae in solutions. (**a**) Netralization, (**b**) Flocculation.

### 2.2. Self-Flocculation Harvesting Efficiency

The harvesting efficiency of self-flocculating microalgae will change with different flocculating conditions. For example, nutrient composition (N, P, $Ca^{2+}$ content) in the culture medium will have a certain influence on the self-flocculation of microalgae cells [15]. Table 1 lists the harvesting efficiency of some self-flocculating algae species. Ahmad et al. [16] found that the addition of $Zn^{2+}$ ions could significantly increase the self-flocculation rate of *Scenedesmus quadricauda* by up to 86.7%. Salim et al. [17] found that the flocculation effect of *Ettlia texensis* could reach 90% in the presence of phosphate for 3 h. Castrillo et al. [18] found that under the same flocculation conditions, the flocculation effect of *Scenedesmus obliquus* was different even if it reached the same pH by adding alkaline substances. The flocculation effect of adding Ca(OH)$_2$ was 10% higher than that of adding NaOH.

**Table 1.** Self-flocculation harvesting efficiency.

| Species | Flocculation Conditions | Flocculation Efficiency | References |
|---|---|---|---|
| *Ettlia texensis* | $PO_4^{3-}$-P = 0.1 g/L | 90%, 3 h | [17] |
| *Chlorococcum* sp. GD | $NO_3^-$-N = 40 mg/L<br>TP = 0.7 mg/L | 84.43%, 3 h | [23] |
| *Chlorella vulgaris* | $Mg^{2+}$ = 20 mg/L<br>$PO_4^{3-}$-P = 1.9 mg/L | pH < 10.5, 0%; pH = 11, 75%;<br>pH $\geq$ 11.5, > 95% | [19] |
| *Scenedesmus quadricauda* | $Zn^{2+}$ = 9.4 g/L | 86.7%<br>pH = 3, 78.4% | [16] |
| *Scenedesmus obliquus* | $Mg^{2+}$ = 7.3 mg/L<br>$PO_4^{3-}$-P = 9.1 mg/L<br>pH = 12 | 90%, $Ca(OH)_2$<br>80%, NaOH | [18] |
| *Desmodesmus* sp. | pH = 7.98<br>6.96 h, 12.37 °C | 94% | [21] |
| *Auxenochlorella Protothecoides* UTEX 2341 | 2 h | 92%, heterotrophic<br>85%, municipal wastewater | [20] |
| *Desmodesmus* sp. PW1 | 2.5 h | >90% | [22] |

Due to the mechanism of self-flocculation, the effect of microalgae self-flocculation will also be affected by the pH of the solution. For example, when Vandamme et al. [19] observed the self-flocculation of *Chlorella*, it was found that no self-flocculation occurred at a pH $\leq$ 10.5; when the pH was adjusted to 11, 75% of the flocculation effect was achieved, while after a pH $\geq$ 11.5, the flocculation effect was significant, reaching above 95%. In addition, the properties of the solution's characteristics will also influence the self-flocculation of microalgal cells. Li Jinyu et al. [20] found that the flocculation efficiency of *Auxenochlorella protothecoides* UTEX 2341 algae species in different sewage types is also different, which was 92% under heterotrophic conditions and 85% under an urban wastewater culture. Zhang et al. [21] optimized the key parameters (pH, temperature and time) of the self-flocculation of *Desmodesmus* sp., which reached a maximum flocculation efficiency of 94% under optimal conditions. Some self-flocculating algae species not only have good self-flocculation ability but can also remove nitrogen and phosphorus from wastewater. For example, *Desmodesmus* sp. PW1 [22] achieved a self-flocculation efficiency greater than 90% within 2.5 h, while the removal of total nitrogen and phosphorus reached 65.3% and 83.5%, respectively. In another study, *Chlorococcum* sp. GD [23] had a flocculation efficiency of 84.43% within 3 h, and the efficiency of nitrogen and phosphorus removal reached 66.51% and 74.19%, respectively.

## 3. Self-Flocculation Affecting Factors

In order to potentially apply the self-flocculation of microalgae in practice, it is not only necessary to study the mechanism of cell self-flocculation but also important to determine the influences of various factors on this process because the flocculation efficiency of these microalgae is affected by many factors. The mechanism of self-flocculation determines its influencing factors. For self-flocculation induced by alkalinity, the fundamental factors are the degree to which the pH can be raised by microalgae photosynthesis and the types and concentrations of positively charged precipitates that can be formed. For self-flocculation induced by EPS, the influencing factors are more complex, theoretically including all the factors affecting the production and composition of EPS.

### 3.1. Species

The self-flocculating strains are selective, and the flocculating conditions and effects will vary with the target algae species, no matter whether the self-flocculation is based on alkalinity or EPS. Based on the formation of self-flocculated algae species by EPS, Salim et al. [24] studied the flocculation effect of four self-flocculating microalgae on *Neochloris oleoabundans*. The results showed that the flocculation effect of *Tetraselmis suecica* was up to 70%, while *Scenedesmus obliquus* reached 30% and the worst effect of

*Ankistrodesmus falcatus* was up to 20%. This may be caused by the differences in surface characteristics and physiological characteristics of algal cells. For example, algal cells with higher electronegative characteristics need more positively charged precipitates, and multicellular and large filamentous algal species are easier to be flocculated and settled than single-cell algal species. In this respect, it is necessary to comprehensively consider the sewage purification capacity of microalgae species and the utilization value of algal cells and select the algal species that are easily flocculated and settled.

### 3.2. Substrates

Because the growth and metabolism of self-flocculating microalgae requires carbon sources, the culture medium generally contains sugars. However, these sugars may also have adverse effects on cell self-flocculation. Studies have shown that organic carbon sources promote microalgae to produce algal organic matter (AOM), and algae organic matters can interfere with the flocculation of polyvalent metal salts and affect the flocculation effect [25]. The lack of nitrogen and phosphorus and other nutrient elements will stimulate microalgae to produce EPS [26,27], which is consistent with bacteria, fungi and other microorganisms. Based on this, high-density culture can be used in operation to obtain a low F/M (food/microorganism, which represents the amount of organic matter per unit weight of activated sludge per unit time) value to naturally form the substrate-limited conditions.

### 3.3. pH

In general, under acidic conditions, no obvious flocculation and almost no sedimentation were observed in microalgal cells. In order to achieve a better self-flocculation effect, the pH of algal fluid needs to be in the alkaline range. Vandamme et al. [19] found that the self-flocculation effect of *Chlorella* increased with the increase of pH. Microalgae such as *Scenedesmus obliquus* and *Limnothrix redekei* can achieve self-flocculation at a higher pH value during culturing. When some microalgae cannot reach a higher pH, they can be induced for self-flocculation by artificially adding alkaline substances such as $Ca(OH)_2$ and NaOH. However, the addition of alkaline substances increases the cost of the microalgae self-flocculation process. However, as more and more flocculating microalgae have been found, some strains show flocculation under acidic conditions. For example, *Scenedesmus quadricauda* [16] reached the maximum flocculation efficiency of 78.4% under pH = 3. Liu et al. [28] developed a flocculation method to induce microalgae self-flocculation by lowering the pH just below the isoelectric point, and the flocculation efficiency was much higher than that of flocculation only by lowering the pH. This phenomenon does not belong to the mechanism of self-flocculation induced by a high pH, which can be explained as the secretion of substances with flocculating activity by microalgae, which causes dispersed algal cells to gather and form flocs.

### 3.4. Temperature

Temperature is a sensitive factor affecting the metabolic activities of microalgae cells, and too high or too low a temperature will have adverse effects on the growth of microalgae. Studies have shown that the optimum temperature range for the growth of most microalgae was 15–30 °C and that the growth of microalgae is slow when the temperature is lower than 15 °C. However, temperatures higher than 35 °C may cause lethal effects on some microalgae [29]. Studies have also shown that temperature has an important effect on the production of EPS by microalgae. High temperature stimulates the formation of EPS, but low temperature inhibits the formation of EPS due to the decrease of cell metabolism. *Botryococcus braunii* almost does not secrete EPS when the temperature is below 23 °C; the optimal temperature is 30–33 °C. *Anabaena* sp. [30] produces little EPS in the range of 30–35 °C, but produces a large amount of EPS only above 40 °C.

### 3.5. Light Intensity

Light is the main energy source for the growth and reproduction of microalgae, which has an important effect on the spontaneous flocculation induced by a high pH and EPS. First, the intensity of light directly determines the degree of photosynthesis of microalgae. The stronger the light, the more sufficient the photosynthesis. The more thorough the inorganic carbon consumption in water, the higher the pH rises, which is more conducive to the occurrence of high pH-induced self-flocculation. However, there is an optimal range of light time and intensity, and growth will not increase or even decrease beyond the saturated light period of algae. Second, light is also a key factor affecting EPS production. Moreno et al. [30] found that after the light intensity increased from 345–460 $\mu$mol/(m$^2$·s), the EPS content of *Anabaena* increased by four times. Therefore, sufficient light is a favorable factor for inducing spontaneous flocculation.

### 3.6. Other Factors

The concentration of specific ions such as $Ca^{2+}$, $Mg^{2+}$ and $PO_4^{3-}$ determines the type of precipitate and the critical pH and has a decisive effect on the spontaneous flocculation induced by a high pH. The higher the concentration of the above ions, the more positively charged precipitates generated. The stronger the electric neutralization ability, the more sufficient the flocculation. Vandamme et al. [19] found that at a pH of 11, calcium and magnesium are expected to precipitate in the form of calcium carbonate, calcium magnesium carbonate, calcium phosphate and magnesium hydroxide. This method depends on the presence of sufficiently high magnesium ion content in the solution ($Mg^{2+} > 0.1$ mM). Therefore, maintaining the above ions at a high level is essential to achieve high pH-induced spontaneous flocculation.

## 4. Comparison with Other Bio-Flocculation Methods

Biological flocculation can be divided into three types: self-flocculation, microorganism and microalgae co-flocculation and microbial flocculant flocculation, which are a biological harvesting technology for harvesting microalgae by using flocculating algae, bacteria, fungi or biological flocculants extracted from microorganisms. Because of its advantages of safety, no secondary pollution, high efficiency, low energy consumption and low cost, self-flocculation is considered one of the most promising harvesting technologies. In order to make a better in-depth study of these three methods and apply them to different microalgae harvesting conditions to obtain an efficient harvesting effect, the mechanisms, advantages and disadvantages of these three flocculation methods are compared in Table 2.

**Table 2.** Comparison of different biological flocculation methods.

| Methods | Mechanism | Advantages | Disadvantages |
|---|---|---|---|
| Self-flocculation | Self-flocculating microalgae cells secrete EPS inducer to sedimentation | No need to add chemical flocculant; high safety; simple operation [31]; low cost and energy consumption; compared with the direct centrifugation method, the energy consumption after self-flocculation can be reduced by 90% [24] | The time of self-flocculation process is long; the harvested microalgae biomass has high water content [32]; the self-flocculating strains have algae strain specificity and their universality needs to be studied |
| Microorganism and microalgae co-flocculation | EPS produced by the metabolic activities of bacteria, fungi, actinomycetes and other microorganisms that induce cell sedimentation | No need to add chemical flocculant; low cost; simple operation; the harvesting effect is good | Microbial contamination may occur |
| Microbial flocculant flocculation | It involves electrostatic adsorption caused by hydroxyl and carboxyl groups in microbial flocculants, or bridging effect enhanced by the combination of this functional group with microalgal cells [33] | Biosafety and degradable; high flocculation efficiency; no secondary pollution | The process of microbial culture and flocculant separation and purification will increase the production cost |

## 5. Perspectives

The collection of such small algal cells from the suspended culture is highly energy-intensive and thus costly for the production of microalgal bioproducts. Thus, it is extremely necessary to explore new harvesting technologies or develop new harvesting agents. Sanchez-Galvis et al. [34] designed a cheap and durable electric flotation system, and the results showed that the harvesting efficiency of algae and cyanobacteria nearly reached 100% in 20 min under 150 rpm using more than ten electrodes. Farid et al. [35] used nano-chitosan to harvest *Nannochloropsis* sp., reducing the consumption of chitosan from 100 to 60 mg/L and increasing the biomass recovery efficiency by 10%. Although the self-flocculation harvesting effect is terrific, there is still a certain gap before the large-scale harvesting of microalgae. Therefore, it needs to be combined with other technologies such as microalgae cultivation in wastewater with self-flocculation harvesting in order to reduce the costs. In addition, co-cultivation of microalgae with microalgae, bacteria or fungi is also expected to efficiently induce self-flocculation, thereby reducing harvesting costs.

### 5.1. Integration of Algal Cultivation in Wastewater with Biomass Harvesting

Microalgae self-flocculation is an ideal potential application technology in microalgae harvesting which integrates cultivation and harvesting without an additional culture system. Self-flocculation does not require the addition of any chemical flocculants without the risk of polluting microalgae cultivation. It is of great significance in the aspects of $CO_2$ emission reduction, sewage treatment and bio-energy production [36]. The growth process of microalgae requires the absorption of a large amount of nitrogen, phosphorus and other nutrient elements, and the cultivation of microalgae in sewage can directly reduce the content of pollutants such as N and P in the effluent [37]. In addition, some microalgae strains with heavy metal adsorption capacity can also remove heavy metals from industrial wastewater [38]. Microalgae can recover inorganic nutrients from wastewater, and their own biomass utilization is equally important. Many species of microalgae have high lipid contents, which can reach 20–50% by dry weight [39]. Using microalgae to treat pollutants in wastewater and harvesting microalgae by self-flocculation to recover biomass energy is an integrated technology with high efficiency and low cost. For example, Chen et al. [22] found that *Desmodesmus* sp. PW1 could not only achieve more than 90% self-flocculation efficiency, but also remove 65.3% of the total nitrogen and 83.5% of the total phosphorus.

Future research should also evaluate the economic feasibility of self-flocculation technology in outdoor large-scale culture to further improve the biological properties of microalgae to produce high lipids through genetic engineering and to optimize the culture conditions of microalgae. Finally, through the integration of various methods and technologies, the reasonable combination of microalgae self-flocculation and sewage treatment for the production of biofuels can be realized so as to achieve the harmony and unity of environmental protection and economic development.

### 5.2. Co-Cultivation of Algae with Algae, Bacteria or Fungi to Assist Flocculation

The self-flocculating algae species can not only flocculate the same algal cells but also induce flocculation of other types of microalgal cells. The flocculation efficiency for harvesting was enhanced to 85.33% in the co-culture of *Desmodesmus* sp. ZFY and *Monoraphidium* sp. QLY-1., which was better than in the monoculture (57.98% and 32.45%) [11]. Moreover, the different ratios of flocculated microalgae and non-flocculated microalgae will also lead to changes in flocculation efficiency. Increasing the ratio will lead to a higher sedimentation rate and recovery rate of non-flocculated microalgae [31].

The flocculation phenomenon of algae and bacteria co-culture systems has attracted the attention of researchers for a long time. In the open sewage treatment system, there is a very complex microbial community structure which correspondently constitutes a complex algal bacteria symbiosis system. Microalgal flocculation can be induced by bacteria and fungi. Bacteria mainly cause algal cell aggregation by directly attaching to the surface of adjacent microalgal cells [40]; bacterial filaments, surface proteins, cell surface

charge and extracellular transparent polymeric particles may also promote algal cell flocculation [41]. Lee et al. [42] found that *Flavobacterium*, *Terrimonas* and *Sphingobacterium* had significant effects on the flocculation activity of *Chlorella vulgaris*. Even in the presence of flocculants and a higher pH, axenic *Chlorella vulgaris* was not able to be flocculated. Fungi mainly cause microalgae flocculation through hyphae [43]. The cell aggregation ability of filamentous fungal cells is mainly due to the production of hydrophobic proteins. The flocculation efficiency of *Chlorella sorokiniana* by co-culture with the filamentous fungus *Isaria fumosorosea* reached 100% within 48 h [44].

However, the main problem related to the co-cultivation mode is that the algae culture requires a high pH, and the addition of fungal inoculum and glucose required for its growth makes the system's pH become acidic. There is a contradiction between the two for the suitable range of pH and a long cultivation time of 24 to 48 h for algae will also inhibit the growth of fungi [45]. In addition, little is known about the mechanisms of algal fungus co-culture flocculation and the key parameters that control the flocculation. In order to develop fungus-assisted algal flocculation in the most effective way, more attention is needed to these aspects. At the same time, appropriate flocculating microorganisms should be selected to reduce pollution, and the activity of microorganisms in algae culture should be further studied to optimize the culture conditions so as to determine whether it can be used for large-scale microalgae harvesting. Therefore, in the future, a variety of means should be used to improve the flocculation efficiency and then potentially realize the industrial application of microalgae in energy.

## 6. Conclusions

Microalgae are a kind of biomass energy source with wide application prospects. The microalgae industry can not only be used as sewage treatment technology but also as biomass energy to relieve the energy shortage, which is of great significance for the sustainable development of the world economy. However, the production of algal biofuel is still challenging due to the limitations of technical economy and the lack of knowledge on algal biological properties. Therefore, it is very important to develop an efficient, pollution-free and low-cost microalgae harvesting technology. Among the existing microalgae harvesting technologies, self-flocculation technology has great development potential. This article has expounded the flocculation mechanism and influencing factors of self-flocculating algae species and can hopefully provide a reference for the follow-up research of algae self-flocculation technology.

**Author Contributions:** Conceptualization, L.Z. and J.H.; methodology, T.L.; validation, L.Z., J.H. and T.L.; writing—original draft preparation, T.L.; writing—review and editing, L.Z.; supervision, J.H. All authors have read and agreed to the published version of the manuscript.

**Funding:** This research was funded by the National Key R&D program of China (2019YFD1101300; 2019YFC1803405).

**Institutional Review Board Statement:** Not applicable.

**Informed Consent Statement:** Not applicable.

**Data Availability Statement:** Data available in a publicly accessible repository. The data presented in this study are openly available in Web of Science.

**Acknowledgments:** This work was supported by the National Key R&D program of China (2019YFD1101300; 2019YFC1803405).

**Conflicts of Interest:** The authors declare no conflict of interest.

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
