# Peer review of "Self-Flocculation as an Efficient Method to Harvest Microalgae: A Mini-Review"

_water, doi:10.3390/w13182585_

Round 1

Reviewer 1 Report

A nice mini-review

1) typos:

lines 8-10: Affiliation and email not correct
lines 13-14: "widely" used twice, word repetition
line 33: increased.. two dots
line 58: CO2 subscript
lines 110, 111: CO2 subscript (HCO3-), super- and subscript
line 123: Ca2+ superscript
line 126: Zn2+ superscript
line 131: Ca(OH)2 subscript
line 151, Table 1: all super- and subscripts to be corrected.
line 164: "The self-flocculating strains are strain-specific"
What does this sentence mean? strains are strain-specific?

2)

line 177: microalgae to produce AOM,--> explain acronym in brackets
line 181: F/M-->explain acronym in brackets

3) Please Elaborate on Batch processes vs. continuous processes.

4) Microalgae can also be used for other application than biofuels, e.g. feed and Food.

5) What is the benefit of the technology in terms of energy efficiency? Compared to e.g. centrifuation.

Reviewer 2 Report

Manuscript Number: water-1359921

Self-Flocculation as an Efficient Method to Harvest Microalgae: A Mini-Review.

The present work shows in a concisely way the different mechanisms of auto-flocculation that take place in the flocculation of microalgae cultures. It exposes its advantages and disadvantages and delves into the parameters by which they are influenced. It has seemed to me a work of great value that requires a long and extensive bibliographic search. It is a starting point for those researchers entering the world of microalgae cultivation.

I think the work is interesting and it should be accepted after some minor revisions.

COMMENTS:

If in line 48, it is referenced as follows: Shirazi et al. [2], in line 84 it should be the same; Sukenik et al. [8].

Section 1.1 does not include the objective of the work, so the authors should include it or modify the name of the section.

In lines 107, 157 and 166, alkaline must be replaced by alkalinity.

Section 3.1 could include examples of the different EPS production capacities depending on the type of microalgae, since they only refer to surface loading, which is one of the two mechanisms mentioned for auto-flocculation.

On line 174 there is an erratum, self-flocculating.

Lines 186-189 are identical to lines 135-137.

The reference included in line 196 should be Liu et al.

Line 265 reference is not correct, Chen et al. it does not appear in the bibliography.
